# Parsing neural dynamics with infinite recurrent switching linear dynamical systems

**Victor Geadah[1], The International Brain Laboratory, Jonathan W. Pillow[1,2]**
[1]Program in Applied and Computational Mathematics, Princeton University.
[2]Princeton Neuroscience Institute, Princeton University.
`{victor.geadah, pillow}@princeton.edu, info@internationalbrainlab.org`

## Abstract

Unsupervised methods for dimensionality reduction of neural activity and behavior have provided unprecedented insights into the underpinnings of neural information processing. One popular approach involves the recurrent switching linear dynamical system (rSLDS) model, which describes the latent dynamics of neural spike train data using discrete switches between a finite number of low-dimensional linear dynamical systems. However, a few properties of rSLDS model limit its deployability on trial varying data, such as a fixed number of states over trials, and no latent structure nor organization of states. Here we overcome these limitations by endowing the rSLDS model with a semi-Markov discrete state process, with latent geometry, that captures key properties of stochastic processes over partitions with flexible state cardinality. We leverage partial differential equations (PDE) theory to derive an efficient, semi-parametric formulation for dynamical sufficient statistics to the discrete states. This process, combined with switching dynamics, defines our *infinite recurrent switching linear dynamical system* (irSLDS) model class. We first validate and demonstrate the capabilities of our model on synthetic data. Next, we turn to the analysis of mice electrophysiological data during decision-making, and uncover strong non-stationary processes underlying both within-trial and trial-averaged neural activity.

## 1 Introduction

Parsing behavioral and neural dynamics into simpler alternating states is providing unprecedented insights into the nature of computation in the brain (Wiltschko et al., 2015; Calhoun et al., 2019; Ashwood et al., 2022; Bolkan et al., 2022). Such data, conventionally collected over multiple trials and spanning considerable lengths of time, is often related to latent processes that exhibit complex dynamics across multiple timescales (Cowley et al., 2020; Roy et al., 2021; Nassar et al., 2019; Yu et al., 2009). Learning is one important example of such non-stationarity, but other changes may be present, such as fatigue (Marcora et al., 2009) or arousal (Schriver et al., 2018). However, inferring those latent processes from data, either within or across trials, remains challenging.

The state-space models that perform this segmentation into discrete states commonly revolve around the hidden Markov model (HMM). This model posits an underlying, hidden, discrete Markov chain, with a different observation model for each state giving rise to the data. Each discrete state can also be extended to capture a different dynamical regime, governing the dynamics of some continuous-space stochastic process. The resulting *switching state-space models* capture activity that alternate between a discrete set of dynamical regimes, and have proved useful in the modeling of complex nonlinear activity (Fox et al., 2010; Smith et al., 2021). A significant step in capturing online dependencies in such models is through *recurrence* (Linderman et al., 2016; Zoltowski et al., 2020), where the continuous dynamical variables govern switches between the discrete states. Such models have been providing powerful insights into the time-dependence of those processes (e.g. Glaser et al. (2020)), but suffer from training challenges due to the fixed cardinality of the discrete states and a lack of geometry over states.

Indeed, traditional HMMs, and models built upon them such as switching state-space models, use a fixed number of discrete states. This fixed cardinality is typically determined by cross-validation, a

procedure that can prove computationally expensive. Furthermore, it does not encourage the use of fewer discrete states on any subset of trials or sessions, so that they can be interpreted and determined as needed. To overcome these limitations, multiple avenues have considered expending HMMs with flexible state cardinality (Beal et al., 2001; Fox et al., 2011). These models revolve around the Hierarchical Dirichlet Process (HDP), which places a flexible non-parametric prior over the transition probabilities between the (theoretically infinite) states. Unfortunately, while the HDP and its generalizations can capture state dependence and persistence (Teh et al., 2006; Fox et al., 2008), in their standard formulation they do not allow for recurrence and other dependencies between the discrete states allocations.

Furthermore, HMMs do not have any *a priori* state geometry. In particular, the discrete states are *permutation invariant*, such that the model is equivalent under any permutation in the label ordering of the discrete states (see schematic in Appendix Fig. 4**B**). This induces non-identifiability and limits the interpretability of the inferred latent discrete states. While recent work has focused on variability in discrete dynamical regimes (Nassar et al., 2019; Linderman et al., 2019; Costacurta et al., 2022) yielding impressive results in terms of flexibility, most of these can be cast as "local" variations ; they consider perturbations or parametrized variations of the parameters associated with each discrete states. To our understanding, none specifically tackle the geometric nature of the discrete states space.

In this work, we use partial differential equations (PDE) theory to develop an alternate prior over the discrete state transitions in recurrent switching state-space models. Our formulation allows us to (1) capture important properties of the stochastic processes for flexible state cardinality, while supporting the use of recurrent connections, (2) induce a geometry over the discrete states, and (3) induce semi-Markovian discrete states dynamics to capture more intricate temporal dependencies. After reviewing relevant background in Section §2, we present in Section §3 our model, the *infinite recurrent switching linear dynamical system* (irSLDS), and provide details on its generative formalism and inference procedure. In Section §4, we first validate the model on synthetic data and showcase its properties, before turning to mice electro-physiology data from the International Brain Laboratory (IBL). The PDE prior is defined by less parameters than the traditional HMM-based transitions model, yet we show that it maintains or even outperform the traditional model while offering an interpretable structure that is amenable to uncovering trial-varying structure.

## 2 BACKGROUND

We review in this section the key models considered in this work. We consider time-stamped data $\{(t_n, \boldsymbol{y}_n)\}_{n=1}^T$ with time steps $t_n$ and data $\boldsymbol{y}_n \in \mathbb{R}^M$.

### 2.1 SWITCHING LINEAR DYNAMICAL SYSTEMS

*Hidden Markov Models* (HMM) posit an underlying discrete-state Markov chain with states $z_n \in \{1, \ldots, K\}$, and conditionally independent observations $p(\boldsymbol{x}_n | z_n)$. *Auto-regressive HMMs* (AR-HMM) extend this framework by endowing the observations $\boldsymbol{x}_n \in \mathbb{R}^D$ with linear dynamics, dependent on the discrete state $z_n$. The generative model reads as

$$z_{n+1} \sim P(z_{n+1} | z_n, n, \boldsymbol{x}_n) \tag{1}$$

$$\boldsymbol{x}_{n+1} \sim \mathcal{N}\left(A^{(z_n)}\boldsymbol{x}_n + a^{(z_n)}, \Sigma_x\right) \tag{2}$$

at time $n \in \{0, \ldots, T\}$, with a different set of dynamics $A^{(z_n)} \in \mathbb{R}^{D \times D}$ and bias $a^{(z_n)} \in \mathbb{R}^D$ for each discrete state $z_n$, and covariance $\Sigma_x$. The switching process in (eq. 1) is described by a transition matrix, written in full generality, but is usually taken to be time- ($n$) and observation ($\boldsymbol{x}_n$) independent such that $P(z_{n+1} = j \mid z_n = i) = A_{ij}$ for some transition matrix $A$.

In this work, we are interested in leveraging those models to study possibly high-dimensional data $y_n \in \mathbb{R}^M$, $M \geq D$, such as spike train data. We thus turn to *Switching Linear Dynamical Systems* (SLDS) models, which model the AR-HMM observations $\boldsymbol{x}_n$ as a (low-dimensional) representation of the data, and model the data $\boldsymbol{y}_{1:T}$ as conditionally independent linear Gaussian emissions

$$\boldsymbol{y}_n \sim \mathcal{N}\left(C\boldsymbol{x}_n + c, \Sigma_y\right) \tag{3}$$

for $n \in \{0, \ldots, T\}$, with decoding weights $C \in \mathbb{R}^{M \times D}$, $c \in \mathbb{R}^M$, and emission covariance $\Sigma_y$. Finally, inputs $\boldsymbol{u}_n$, e.g. the time steps $t_n$, may be linearly encoded in the continuous dynamics (eq. 2)

or directly along with the emissions in (eq. 3). Following Fox et al. (2010), we refer to AR-HMM and SLDS models as instances of *Markov switching processes*, or generally *switching state-space models*.

A vital augmentation of Markov switching processes is to allow observations or internal states to guide the switches in discrete states. Linderman et al. (2016) introduced such a $x_n \rightarrow z_{n+1}$ dependency (blue arrows in Fig. 1**E**) in the SLDS model class, coining it *recurrence*. Now in the *recurrent SLDS* (rSLDS), in place of the general equation in eq. (1), the continuous states $x_n$ guide the switching through

$$z_{n+1} \mid z_n, x_n \propto R^{(z_n)} x_n + r^{(z_n)}$$

for recurrent encoding matrices $R^{(\cdot)} \in \mathbb{R}^{K \times D}$ and biases $r^{(\cdot)} \in \mathbb{R}^K$. This is the central model we consider in this work—see Fig. 1**E** for the graphical model.

## 2.2 DISTANCE DEPENDENT CHINESE RESTAURANT PROCESS

The switching process in (eq. 1) sets the discrete dynamical modes of the continuous dynamics. Modeling and inferring trajectories under this process amounts to the problem of clustering the dynamics in a discrete set of dynamical modes. As stated in the introduction, we wish to infer the number of states $K$ from the data, contrary to the formulation above. Dirichlet Processes (DP) are a classical tool in this case, as they provide an infinite random measure over cluster partitions (Ferguson, 1973; Antoniak, 1974). They can be described by the Chinese Restaurant Process (CRP, a form of Pólya urn process, see primer in Appendix §A), which captures the self-reinforcing and non-parametric structure of the prior. A key limitation is that points under the CRP are *exchangeable* (Blei & Frazier, 2011) (see schematic in Appendix Fig. 4**A**)—under any permutation of the ordering, the probability of a configuration is the same. The purpose of recurrence is to actively control the dynamical mode transition, which fundamentally breaks exchangeability.

We seek a stochastic process over discrete state partitions that will allow us to capture recurrent (i.e. online) dependencies on the data or latent states. To this end, we turn to the *distance dependent CRP* (dist-CRP) from Blei & Frazier (2011). This variant of the CRP breaks exchangeability by associating each time step to another, and *then* perform clustering based on pairwise assignments. Specifically, at time step $n$, this process assigns time step $i \in [n]$ with $c_i \in [n]$ following

$$p(c_i = j | D, \alpha, \beta) \propto \begin{cases} f(D_{ij}; \beta) & \text{if } i \neq j \\ \alpha & \text{else} \end{cases} \tag{4}$$

with distance matrix $D_{ij}$, decay function $f(\cdot; \beta)$ and decay parameter $\beta > 0$. As we consider time-stamped data, we consider the time difference $D_{ij} = t_i - t_j$ for $i \geq j$, and enforce that no step is assigned with future steps by setting $f(D_{ij}) = 0$ if $i < j$. Finally, we use the entire history $c_{:n} = \{c_{:n-1}, c_n\}$ of pairwise assignments to perform clustering: if $c_i = c_j$, then $i$ and $j$ are in the same cluster $z_i$. Applying this to the history $c_{:n}$, we get $z_{:n}$, thereby setting $z_n$. The graphical model for this process is illustrated in Fig. 1**A**.

## 3 INFINITE RECURRENT SWITCHING LINEAR DYNAMICAL SYSTEMS

Our technical contribution consists in using key properties of the dist-CRP to guide the switching process in the rSLDS in a manner that supports efficient inference and generation. Section §3.1 presents the fallouts of a naive combination, providing details on how conventional methods relying on Pólya-gamma augmentation yield intractable Bayesian inference in this case. Section §3.2 motivates our alternative approach to modeling based on sufficient statistics and partial differential equations (PDEs), and highlights key non-exchangeable and geometrical properties of our model. Finally, Section §3.3 provides implementation details for both generation and inference.

### 3.1 PÓLYA-GAMMA AUGMENTATION YIELDS INTRACTABLE INFERENCE

Pólya-gamma augmentation (Polson et al., 2013; Linderman et al., 2015) is a powerful augmentation strategy that handles non-Gaussian factors by introducing additional latent variables to obtain joint Gaussian likelihoods. It was used in the original formulation of the rSLDS model by Linderman et al. (2016) to allow tractable inference with message passing. We show in Appendix §C.1 that

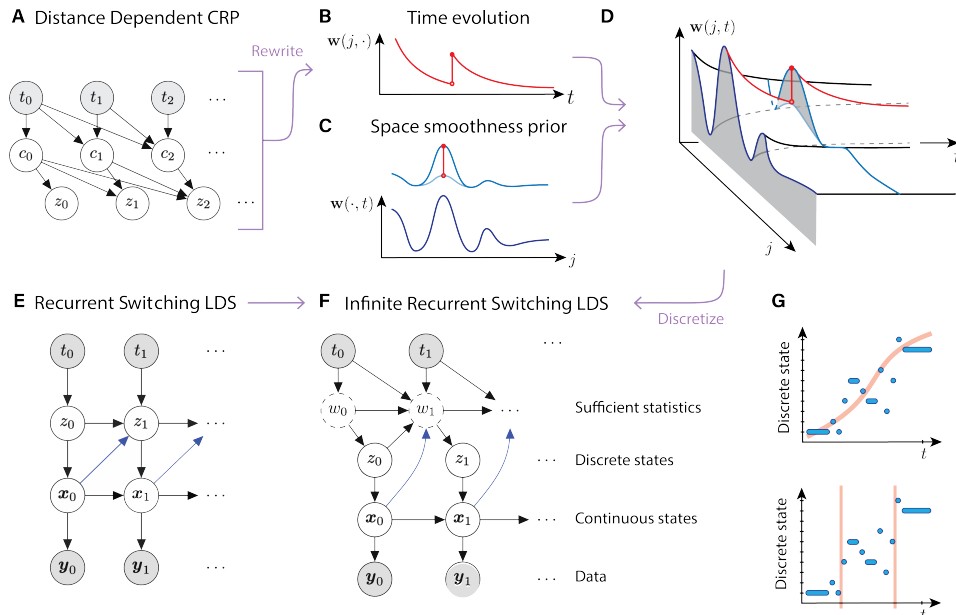

Figure 1: *Modeling details*. (**A**) Graphical model for the distance dependent CRP (dist-CRP) in its conventional form. (**B**) The generative process of the dist-CRP can be summarized with a sufficient statistic $\mathbf{w}(j,t)$ for state $j$ at time $t$, showcasing a typical choice-driven exponential decay. (**C**) We induce a geometry over the state space $j \in \mathcal{J}$, and add a spatial smoothness prior. (top) Each time a state is chosen (red bar), its probability is increased for future steps and so is the probability of nearby states. (**D**) Modeling the influence function $\mathbf{w}(j,t)$ as a solution to the heat equation. (**E**) Graphical model for the recurrent switching linear dynamical system (rSLDS). The recurrence is highlighted by the blue arrows. (**F**) The *infinite rSLDS* model combines the heat-equation generalization of the dist-CRP for the discrete states with the rSLDS switching linear dynamics emission process. (**G**) The irSLDS discrete state prior better supports geometric analyses (top) and a variable number of occupied states (bottom).

for appropriate choices of decay function $f$, we can also leverage Pólya-gamma augmentation to similarly handle the non-Gaussian factors emerging from recurrence. However, the resulting Gaussian augmentation grows linearly for each step $n$, making it computationally inefficient. Luckily as we'll see below, we can use sufficient statistics and recurrent dynamics to circumvent this problem.

## 3.2 PDE PRIOR FOR EFFICIENT PARAMETRIZATION AND NON-EXCHANGEABILITY

To remedy the challenges arising out of a naive combination of the original dist-CRP and SLDS models, we rely on the following sufficient statistic to express the cluster allocations

$$p(z_n = j | \mathbf{c}_{:n}, \alpha, \beta) = p(z_n = j | \mathbf{w}(\cdot, t_n), \alpha, \beta) \propto \begin{cases} \mathbf{w}(j, t_n; \beta) & \text{if } j \text{ in history} \\ \alpha & \text{else} \end{cases} \quad (5)$$

where $\mathbf{w}(j,t)$ denotes the *influence function* of state $j$ at time $t$

$$\mathbf{w} : \mathcal{J} \times \mathbb{R}_+ \to \mathbb{R}_+, \quad \mathbf{w}(j,t) = \sum_{\{i \,:\, t_i \leq t, z_i = j\}} f(t - t_i; \beta)$$

with distance function $f(\cdot; \beta)$ and decay parameter $\beta > 0$. Using this sufficient statistic makes the entire process Markovian. Setting $f(x; \beta) = \kappa \exp(-\beta x)$ to be an exponentially decreasing function, notice that we can rewrite $\mathbf{w}(j,t)$ above as a solution to the continuous time ODE

$$\frac{\partial}{\partial t} \mathbf{w}(j,t) = -\beta \mathbf{w}(j,t) + \kappa \mathbf{1}_{\{z_n = j\}}, \quad \forall j \in \mathcal{J} \quad (6)$$

with càdlàg trajectories $\mathbf{w}(j,t)$ in time $t$. Each time $t_n$ a discrete state $z_n = j$ is chosen, its influence function is bumped by $\kappa > 0$, increasing the weight of this state for future time steps (see Fig. 1**B**). This has a similar effect as the "stickiness" parameter in the sticky HDP-HMM (Fox et al., 2008).

In practice, we need to evolve in parallel the influence functions $\mathbf{w}$ for each state $j$. For any fixed $t$, we make the modeling choice of relating these states to one another by using a smoothness prior (see Fig. 1**C**). Inspired by the temporal dynamics in (eq. 6), we model the sufficient statistic to evolve according to a partial differential equation (PDE), the *heat equation* (see Fig. 1)

$$\frac{\partial}{\partial t}\mathbf{w}(j,t) = \gamma\frac{\partial^2}{\partial j^2}\mathbf{w}(j,t) \tag{7}$$

with diffusion coefficient $\gamma > 0$ and with initial conditions for $t \geq t_n$

$$\mathbf{w}(\cdot,t_n) = \lim_{t \uparrow t_n}\mathbf{w}(\cdot,t) + \kappa\mathbf{1}_{z_n}, \quad \mathbf{w}(\cdot,0) = \mathbf{0}$$

over the state space $j \in \mathcal{J}$. While many alternate priors could offer a form of spatial smoothness with similar temporal dynamics, we choose the heat-equation for several reasons:

1. The temporal dynamics encompass the dist-CRP. Indeed, by standard separation of variables argument, the temporal component of a solution $\mathbf{w}$ to (eq. 7) exhibits the exponential decay desired in (eq. 6) for some appropriate $\beta > 0$.

2. The spatial dynamics offer an intuitive notion of diffusion of probabilities. Each time a discrete state $j$ is chosen, "nearby" states also become more probable at the next time step (see Fig. 1**C** top).

3. The heat-equation as a differential operator on functions possesses numerous linearity qualities, which makes general numerical behavior and approximations favorable.

We stress the meaning of "nearby": in models based around an HMM, the states do not have an inherent geometry[1]. In comparison, here discrete states can be interpreted as points on a continuous state space $\mathcal{J}$. We can restrain this space to a discrete number of states, and we discuss how to do so in practice in the next subsection.

## 3.3 IMPLEMENTATION AND INFERENCE IN INFINITE RECURRENT SLDS

We proceed by describing some of the implementation implications of the PDE prior presented in the previous section. We assume from now on regularly-sampled data in time, of difference $\Delta t$, and drop the inclusion of the time data $t_n$ to only consider the emissions data $\boldsymbol{y}_n$. The code for this work builds on the SSM package (Linderman et al., 2020), and we present in Appendix C.2 the relevant modules.

**PDE prior implementation with finite difference methods**  We use finite difference methods to implement a sufficient statistic $\mathbf{w}$ solving the heat-equation prior. Let $[\boldsymbol{w}_{n+1}]_{j+1} := \mathbf{w}(j + \Delta j, (n + 1)\Delta t)$ be our discrete approximation. We use a forward difference of the time derivative and central difference approximation to the second order spatial partial derivative to obtain the solution

$$\boldsymbol{w}_{n+1} = U\boldsymbol{w}_n, \quad U = \text{tridiag}(\beta, 1 - 2\beta, \beta) \in \mathbb{R}^{K \times K}$$

where $\beta = \gamma\frac{\Delta t}{\Delta j^2}$. We impose $\Delta t \leq \frac{(\Delta j)^2}{4\gamma}$ as a general requirement for stability, and let $\Delta j$ be adjusted accordingly given $\gamma$ (model parameter) and $\Delta t$ (data parameter). Inputs can be added to drive the system, including (1) the desired $\kappa\mathbf{1}_{z_n=j}$ adding self-reinforcement to the system, and (2) the past internal states encoded by a module of weight $R \in \mathbb{R}^{K \times D}$ and bias $r \in \mathbb{R}^K$ for recurrence. The dimension $K$ of $\boldsymbol{w}_n \in \mathbb{R}^K$ is a fixed hyperparameter. Importantly, however, the heat equation dynamics capture properties of the dist-CRP, such that $K$ rather acts as an upper bound on the parsimonious number of states used per trajectory. In all, the dynamics of $\boldsymbol{w}_n$ follow

$$\boldsymbol{w}_{n+1} \mid \{\boldsymbol{w}_n, \boldsymbol{x}_n, z_n = i\} = U\boldsymbol{w}_n + \kappa\mathbf{1}_i + R\boldsymbol{x}_n + r \tag{8}$$

with parameters of diffusion and decay $\gamma \propto \beta > 0$ and self-reinforcement $\kappa \in \mathbb{R}$. The $j$-th entry of the vector $\boldsymbol{w}_{n+1}$ in (eq. 8) defines, up to a necessary row-normalization factor, the $ij$-th entry of our transition matrix $[W_{n+1}]_{ij}$. Dynamical modes $z_n$ are then sampled according to (eq. 5), with final random transition parameter $\alpha > 0$. Now a few scalar parameters $\{\alpha, \gamma, \kappa\}$ govern the discrete state $z_n$ process, contrasting with the $K \times K$ matrix in the rSLDS. In all eqs. (8, 5) with the switching continuous dynamics in (eq. 2) and emissions in (eq. 3) together define the *infinite recurrent SLDS* (irSLDS, see graphical model in Fig. 1**F**). We refer to Appendix §D.3 for a study of the behavior of the model as we scale $K$, the influence of $\{\alpha, \gamma, \kappa\}$, and the values used for the latter.

---

[1]One can consider spectral clustering from the graph Laplacian defined from the transition matrix. This however remains a transformation, and the underlying states originally have no geometry.

**Inference with variational Laplace-EM**    The added sufficient statistic $\boldsymbol{w}_n$ in (eq. 8) is deterministically determined given $z_{n-1}$ and $\boldsymbol{x}_{n-1}$, thus adds no component to the posterior inference. We follow prior work and perform inference in this model using variational Laplace-EM from Zoltowski et al. (2020). As an overview, we posit the same structured mean-field approximation $q(\boldsymbol{z}_{1:T})q(\boldsymbol{x}_{1:T})$ to the states posterior. For our continuous states posterior, we use a Laplace approximation around the most likely path $\hat{\boldsymbol{x}}_{1:T}$ (Paninski et al., 2009). Given a continuous states trajectory under this posterior, the transition matrices $W_{1:T}$ can be computed to define our model joint $p(\boldsymbol{y}_{1:T}, \boldsymbol{z}_{1:T}, \boldsymbol{x}_{1:T})$. From there, the discrete state posterior approximation is found by locally maximizing the expected model joint under the continuous state posterior. Generally, conditioned on samples $\boldsymbol{x}_{1:T}$, the factor graph for our $\boldsymbol{z}$ posterior is equivalent to that of an heterogeneous HMM. Common tools can then be used, such as importantly the Viterbi algorithm to obtain the most likely discrete state sequence $\hat{\boldsymbol{z}}_{1:T}$ conditioned on $\hat{\boldsymbol{x}}_{1:T}$, which we will consider below.

**Discrete state geometry**    The support of $\boldsymbol{w}_n$ is taken to be the set $\{\Delta_j, 2\Delta_j, \ldots, K\Delta_j\}$, which lives on the latent geometry $\mathcal{J}$ used in our PDE formalism. In this work we consider one possible way to leverage this geometry: to define intermediate states. Indeed, the continuity of $\mathcal{J}$ allows for interpolation between states. From any distribution on our discrete states $z_n \in \{1, \ldots, K\}$, or continuous interpolation thereof, one can define different statistics of interest such as the mean or the mode. Such statistics can take continuous values in $\mathcal{J}$, and are ill-defined for traditional HMM-based models due to permutation invariance. In particular, given the most likely sequence $\hat{\boldsymbol{z}}_{1:T}$ and our posterior $q(z_{n+1} \mid z_n, \boldsymbol{x}_n)$ with interpolation function $\tilde{q}$ over $\mathcal{J}$, we will consider the *interpolated sequence* $\tilde{\boldsymbol{z}}_{1:T}$ as the interpolated posterior modes

$$\tilde{z}_n = \underset{z \in \mathcal{J}}{\arg\max}\, \tilde{q}(z_n = z \mid \hat{z}_{n-1}, \hat{\boldsymbol{x}}_{n-1}), \quad n \in \{1, \ldots, T\}, \tilde{z}_0 = \arg\max \tilde{q}(z_0) \qquad (9)$$

**Parallelizing transition matrix dynamics with scans**    In both generation and inference, the core computational difference in terms of complexity between the irSLDS and the (r)SLDS models is that the transition matrix $W_n$ for the discrete states $z_n$ now possesses its own recurrent dynamics (from eq. (8), see Appendix eq. (13)). While the (r)SLDS models also have time-varying transition matrices, they do not have such a recurrent term. To illustrate the difference, considering eq. (8) for row $i$ of the transition matrix $W_n$, the equivalent notation for the rSLDS would be

$$\boldsymbol{w}_{n+1} \mid \{\boldsymbol{x}_n, z_n\} = [\boldsymbol{w}_0]_{z_n} + R\boldsymbol{x}_n + r.$$

Computing the $z_n$ transition matrices in the rSLDS thus amounts to the cost of encoding the continuous state sequence $\boldsymbol{x}_{1:T}$, which can be distributed over different processors. In comparison, we are forced to compute sequentially the transition matrices statistics $W_n$, naively taking $\mathcal{O}(K^3T)$ operations in addition to the sequential encoding of $\boldsymbol{x}_{1:T}$. Fortunately, notice that the dynamics (Appendix eq. 13) are linear. In this case, as presented by Blelloch (1990), computing the transition matrices can be cast as a *scan* operation (see Smith et al. (2023) for a recent application of this concept). With this, we can efficiently parallelize our transition matrix dynamics and reduce the computation to $\mathcal{O}(K^3(T/L + \log L))$ operations over $L$ processors, resulting in similar time complexity as the rSLDS model. We refer to Appendix C.2 for code and more details on our use of the parallel scan.

## 4    EXPERIMENTS

We train models by maximizing the Evidence Lower Bound (ELBO) using variational Laplace-EM from Zoltowski et al. (2020). We compare model performance using the marginal log likelihood (LL)

$$\log p(\tilde{\boldsymbol{y}}_{1:T}) = \int p(\tilde{\boldsymbol{y}}_{1:T}|\boldsymbol{x}_{1:T}, \boldsymbol{z}_{1:T})p(\boldsymbol{x}_{1:T}, \boldsymbol{z}_{1:T})d\boldsymbol{x}_{1:T}d\boldsymbol{z}_{1:T},$$

which is the log-probability of held-out test data $\tilde{\boldsymbol{y}}_{1:T}$ under a given model, where test data arises from a 4:1 train-to-test split of the full dataset (see details in Appendix §B). The required integral is high-dimensional and intractable, and we thus resort to sequential Monte Carlo (SMC), also known as particle filtering, to compute it (Del Moral et al., 2006; Kantas et al., 2009).

### 4.1    VALIDATION ON THE SYNTHETIC NASCAR TASK

We consider the synthetic NASCAR task as used by Linderman et al. (2016). In the original task, the true underlying model is a rSLDS with $K = 4$ states, dissecting the 2-D state space in four

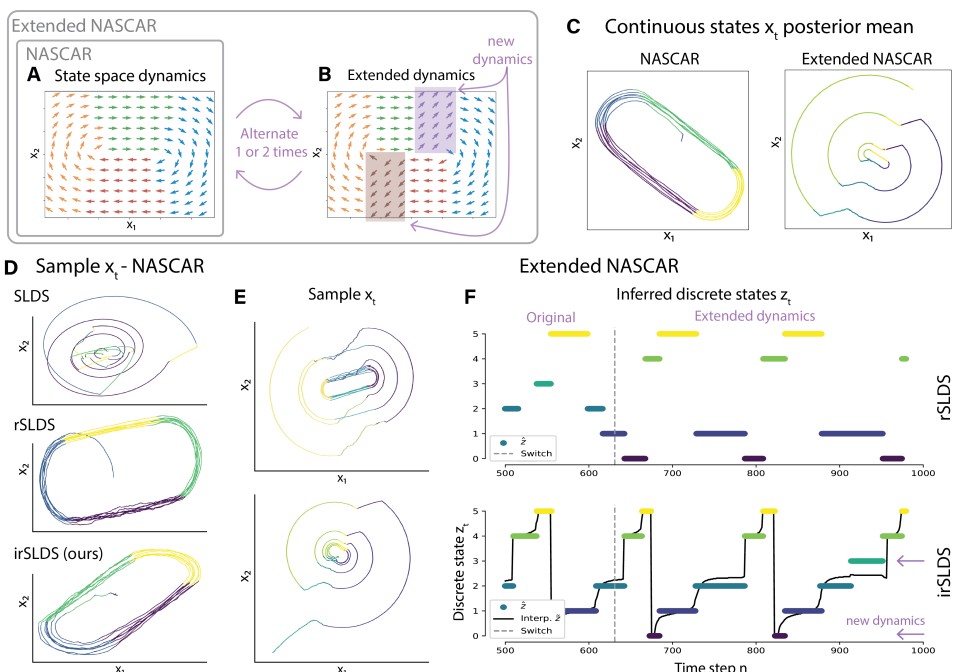

Figure 2: *Synthetic NASCAR experiments.* (**A-B**) True flow field for the NASCAR task in **A**. The extended NASCAR task alternates between the flow field in **A** and the extended flow field in **B**. (**C**) Mean continuous trajectories $\hat{x}_{1:T}$ under the variational Laplace posterior, per task. (**D-E**) Sample continuous latent states $x_{1:T}$ trajectories per model (NASCAR in **D**, extended NASCAR in **E**). (**F**) Most likely discrete state trajectories $\hat{z}_{1:T}$ for the rSLDS (top) and irSLDS (ours, bottom) for one data trajectory in the extended NASCAR task (color scheme matches **E**), along with the interpolated sequence $\tilde{z}_{1:T}$ (eq. 9) in black for the irSLDS.

Table 1: *Performance metrics on NASCAR experiments.* We report the marginal log-likelihood on test data (*Test LL*), evaluated with SMC. Test data-sets consist of $T = 200$ excluded steps in NASCAR, and excluded trajectory of $T = 1000$ steps in extended NASCAR. Column values for $K$ denote value set for the model. We also report the dynamical mean-square error (*Dyn. MSE*) between the sample $x$ state-space flow and true flow field. See Appendix Table 2 for dynamical MSE error bars and number of parameters.

| | NASCAR | | | | Extended NASCAR | |
| | Test LL (↑) | | Dyn. MSE (↓) | | Test LL (↑) | Dyn. MSE (↓) |
| Model | $K = 4$ | $K = 8$ | $K = 4$ | $K = 8$ | $K = 6$ | |
|---|---|---|---|---|---|---|
| SLDS | 1331.35 | 1592.81 | 0.930 | 305.046 | 7353.22 | 316.75 |
| rSLDS | 1628.57 | 1613.53 | 0.010 | 0.011 | 8015.93 | 0.35 |
| irSLDS (ours) | 1656.10 | 1611.45 | 0.015 | 0.011 | 8219.41 | 0.24 |

quadrants with rotational and bias dynamics (see Fig. 2**A**). The resulting continuous latent states $x_n$ resemble car tracks on a NASCAR driving track (see Fig. 2**C**-left for corresponding mean posterior estimates under variational Laplace-EM), and the data is obtained from high-dimensional ($N = 10$) Gaussian emissions of the $x_n$ dynamics. We compare the irSLDS against the rSLDS and SLDS models, presenting results over 5 random initialization seeds.

First, we found both our model, the irSLDS, and the true generative model, the rSLDS, to be similarly able to generate accurate samples of the dynamics (see Fig. 2**D**). In terms of performance on held-out time steps of the trajectory ($T = 200$), both models attained similar accuracy (see Tab. 1). The irSLDS attained the higher mean on the true number of states $K = 4$, and both models performed similarly for the over-complete $K = 8$. The SLDS however did not perform as well, as to be expected—this NASCAR task was introduced to test the inclusion of recurrence in the rSLDS,

compared to the standard SLDS. While the SLDS model attained relatively high test LL (see Tab. 1), it struggled to accurately depict the true dynamics (see Fig. 2**D**-left). We refer to this generating process accuracy as *dynamical accuracy*, which we quantify by reporting the mean squared error (MSE) between the learned dynamics and the true, available, dynamics. We minimize the MSE over reflections and rotations of the learned dynamics field. The SLDS model showed poor dynamical accuracy (see Tab. 1), while the irSLDS matched the accuracy of the true rSLDS. In all, this validates the incorporating of recurrence into our model, and furthermore provides evidence that the PDE prior can mimic the performance and generative capabilities of the fully-trained HMM prior.

The original NASCAR task has a periodic temporal structure (a property unchanged by permutation invariance) with a constant number of required states $K$—both stationary properties. Before turning to our target experimental data, we assess the handling of non-stationary challenges by considering an *extended NASCAR* task. In this new task, the dynamics alternate between the original dynamics (Fig. 2**A**) and a set of new extended dynamics (Fig. 2**B**) where two additional bias dynamics are introduced. This effectively changes the true number of discrete states from $K = 4 \to 6$ during this extended dynamics block. We consider $B \in \{2, 3\}$ blocks of alternating dynamics, each of same expected block lengths $T/B$. In this new task, we found the irSLDS to outperform the rSLDS, in turn outperform the SLDS (Tab. 1). Looking into the learned solutions for the rSLDS and irSLDS models, we observe that the sample trajectories (Fig. 2**E**) in the irSLDS more closely match the posterior mean from data. Furthermore, the irSLDS model correctly identifies two new states after the switch (see Fig. 2**F**-bottom). Interestingly, the posterior ends up attributing these two states to be in between previous states, such that the sequence of discrete states exhibits this traversing trajectory. In contrast, the rSLDS model mistakenly maintains only 4 active states after the switch to the alternate dynamics ( Fig. 2**F**-top). Together, this provides evidence for the performance of the irSLDS in non-stationary tasks and an example on how we can leverage the discrete state geometry of the irSLDS.

## 4.2 The irSLDS uncovers trial-varying structure in neural dynamics regimes

Next, we turn our attention to an electrophysiological dataset from the International Brain Laboratory (IBL), recorded in mice during a sensory-motor decision-making task. In this task, mice reported the location of a sinusoidal grating stimulus by turning a wheel either left or right, with task difficulty controlled by stimulus constrast (Fig. 3**A**) (Laboratory et al., 2021). We consider Neuropixels probe recordings from the "Brainwide map" data release (Laboratory, 2022). For our analyses, we projected spike train data onto the top principal components to obtain firing rates, making it amenable to analysis by state-space models with Gaussian emissions (Fig. 3**B-C**) (further methodological and data details can be found in Appendix §B, including firing-rate and continuous latent dimensions). As the irSLDS directly expands on the rSLDS, we focus on comparing these two models in this section. We picked $K = 8$ discrete states ( see Appendix Tab. 3 for test marginal LL values for $K \in \{2, 4, 8\}$).

First, we found that the models uncovered discrete latent states that switched at task-relevant event times. Indeed, we did not provide the models with task event times such as the stimulus onset ("Stim On") or reward, or behavioral measurements such as the movement onset time ("First movement"). Given that a task event occurred at time step $t$, we plot in Fig. 3**D** the estimated probability of a switch a time-step $t + l$ for a lag $l$ ($p(\text{switch}_{t+l}|\text{event}_t)$), for the irSLDS. We compare this against the baseline probability $p(\text{switch}_t)$, obtained by trial shuffling. The models learned task switches significantly different from chance, capturing a switch in discrete states either preceding or following a relevant task event. We thus posit that the statistical models capture relevant dynamics for the tasks.

Second, we found that the irSLDS uncovered differences in the discrete state distribution over trials. In Fig. 3**E** we plot the number of active states used in the most likely $\hat{z}_{1:T}$ trajectories under the variational posterior. We fit a spline function to this data for both models, determined from the minimal $p$-value of the $F$-test for a comparison against a constant function (computed over various degrees of smoothness and degree). While both models differed significantly from a constant function ($p \ll 0.001$), the irSLDS uncovered larger fluctuations in the number-of-discrete-states-per-trial curve. Fluctuations were much smaller for the rSLDS model, and the spline provided a poorer fit ($R^2$ of 0.16 v.s. 0.46 for irSLDS). This indicates that discrete state distribution over trials fluctuated more dramatically and systematically under the irSLDS model, with substantially more discrete states used in the middle of the session than at the beginning or end. The irSLDS attained a slightly higher log-likelihood on held-out test trials (see Tab. 3), so in all this indicates that those fluctuations are important, and might be missed by traditional HMM-based priors.

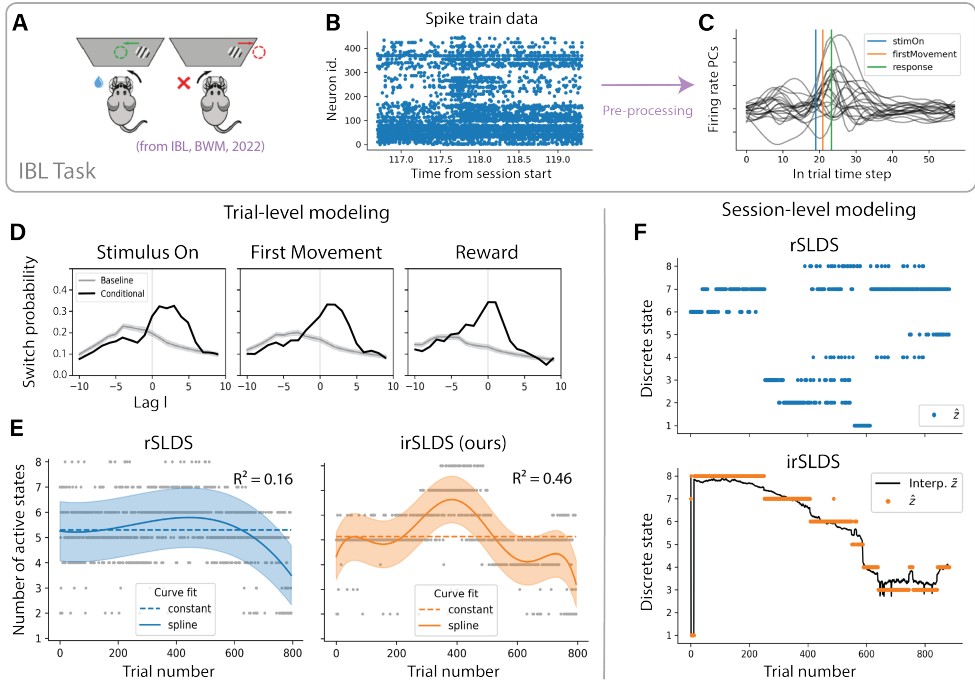

Figure 3: *Results on neural electro-physiology data from the IBL.* (**A**) Experimental paradigm in the IBL task, taken from Laboratory (2022). See text for further details. (**B**) Example spike train data, plotting spike times per neuron, for one trial in one session. (**C**) The trial in **B**, pre-processed with kernel smoothing and PCA, along with relevant trial events. (**D**) Switching probability as a function of the time-step difference (lag $l$) for two key trial events, either conditioning on the event happening at lag 0 ("conditional", black), or without conditioning ("baseline", gray). (**E**) Number of actives states occupied by most likely discrete states trajectories per trial. Curve fits, either constant of spline, are plotted against the number of states per trial in gray. The $R^2$ for the spline fit is indicated top right. (**F**) Most likely discrete states $\hat{z}_{1:N_{\text{trials}}}$ trajectories underlying per-trial average neural responses.

To further investigate the task-induced neural activity, we turn from trial-level modeling to *session-level* modeling. Specifically, we focus on the average firing rate $\boldsymbol{y}_t \in \mathbb{R}^D$, averaging from 200ms before the stimulus onset to the reward, for each trial $n \in [N_{\text{trials}}]$. Fig. 3**F** shows the most likely discrete state sequences $\hat{\boldsymbol{z}}_{1:N_{\text{trials}}}$ under each model. Both models achieve similar training ELBO (see Appendix Tab. 3). We observe that the irSLDS showcases a distinctive slow geometric drift in the discrete states occupied, as observed from $\hat{\boldsymbol{z}}$ and further solidified by the trace of the interpolated $\tilde{\boldsymbol{z}}$ (defined in eq. 9). Finally, we note how the interpolation also indicates the presence of uncertainty between states 3 and 4 at the end of the session. Such conclusions cannot be drawn in the (r)SLDS, as any permutation of the discrete states yields an equivalent visualization of the discrete state process.

## 5 CONCLUSION

In this work, we extend recurrent switching state-space models with an input-driven heat-equation prior over the dynamics of the transition matrix. This results in a semi-Markov discrete state process that capture the behavior of stochastic processes on partitions for a variable number of states per trajectory, as well as inducing a continuous geometry on the discrete states. We show that while this process is defined by less parameters than the traditional HMM models, it still matches or outperforms the original model while providing insights into time-varying processes in data. We showcase the model on a synthetic task, before turning to electrophysiological data from the IBL. The IBL hosts extensive datasets covering multiple repeated- to brain-wide recordings (Laboratory et al., 2022; Laboratory, 2022), and the modeling presented here provides grounds for further investigation into the time-varying processes underlying neural data.

## ACKNOWLEDGMENTS

The authors would like to thank Rich Pang and David Zoltowski for helpful discussions and code suggestions early in the project. VG was supported by doctoral scholarships from the Natural Sciences and Engineering Research Council of Canada (NSERC) [PGSD3-557875-2021] and the Fonds de Recherche du Québec Nature et technologies (FRQNT) [B2X 297667]. JWP was supported by grants from the Simons Foundation (SCGB AWD543027), the NIH (R01EY017366), the NIH BRAIN initiative (NS104899 and 9R01DA056404-04), and the CAREER award (IIS-1150186).

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

**Broader Impact** The modeling here presented is targeted towards enhancing our understanding of neural computation. Analysis of electrophysiological data can have long-term implications for the treatment and understanding of medical treatment and neurological disorders. However, these considerations are far removed for the preliminary analysis and modeling presented, and we foresee no immediate societal consequences of this work.

**Limitations** We highlight a few limitations. First, in practice for our experiments, our implementation of the scan operation for parallel computing, as presented in section §3.3, does not support the automatic differentiation used for our variational inference. We thus only use the scan partially—we refer to section C.2 for more details on the manner. Second, we present early results on ephys data, for a limited number of sessions, probe locations, animals, etc. We do not aim to make general claims about computation in neural systems, but rather highlight the properties of our models in uncovering time-dependent processes. Expanding to the aforementioned data modalities is grounds for rich future work.

**Compute** All experiments were run on an external clusters. For reference, training on a single session for the ephys IBL data might take up 12 hours for a single model, with some multiprocessing. Training a single model for the synthetic NASCAR data can be accomplished on the order of 30 minutes.

## A  BACKGROUND

### A.1  CHINESE RESTAURANT PROCESS

Dirichlet Processes (DP) mixture models provide a powerful random measure over clusterings. We refer to Ferguson (1973); Antoniak (1974) for a measure-theoretic treatment of DPs, and Teh et al. (2006) for a machine-learning overview. They can be alternatively defined through the *Chinese Restaurant Process* (CRP), a process akin to the Pólya Urn process. The analogy goes as follows : upon entering a restaurant, a customer $i$ selects to sit at a table $k$ with probability proportional to the number of people already sat at that table. With some fixed rate $\alpha$, they may decide to start a new table. Put otherwise, for a new customer $i$, its table allocation $z_i$ follows

$$p(z_i = k | z_{:i}, \alpha) \propto \begin{cases} n_k & \text{if } k \leq K \\ \alpha & \text{if } k = K + 1 \end{cases} \tag{10}$$

with $n_k$ the size of cluster $k \in [K]$. In this work, one should think of tables as clusters or *dynamical modes*, and the customers $i$ as time-steps. It can be easily seen that this process induces a joint probability over cluster assignments that is invariant to the order of customers entering. We call this property *exchangeability*. This enforces a strong and limited prior on distributions of partitions that can arise from this model.

## B  TRAINING DETAILS

### B.1  NASCAR TASK

**Data generation** We generate NASCAR track trajectories by instantiating the true model as described, and generating a sample trajectory running for $T = 1000$ steps. We then split the first 800 steps as train data, and keep the last 200 steps as test data (4:1 split). We repeat this procedure for 5 random seeds. For all models, we use $K = 4$, $D = 2$ ($\boldsymbol{x}_t$ dimension) and $M = 10$ (observation $\boldsymbol{y}_t$ dimension).

### B.2  IBL TASK

**Pre-processing** Given spike train data from a given period, we first perform the following pre-processing steps to obtain the firing rates over the period

1. Kernel smoothing against a Gaussian kernel to obtain firing rate graphs. We use a standard-deviation of 100ms if the period is a whole trial, and of 30ms if the period is the Open-Loop period.

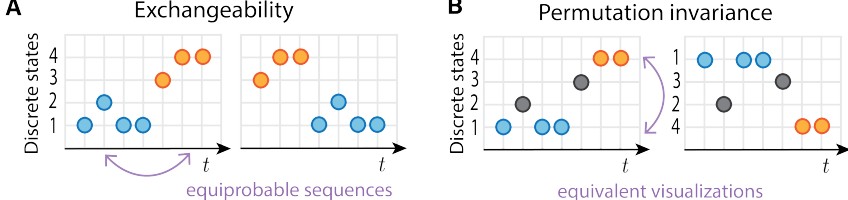

Figure 4: Schematic representation of (**A**) exchangeability, notably in the Chinese Restaurant Process, and (**B**) permutation invariance, notably in HMMs and HMM-based models.

2. The firing rate graphs are evaluated over an evenly space grid over the period interval. The bin size for the grid is determined following Algorithm 1 from Shimazaki & Shinomoto (2007), taking the minimal value, after removing outliers, over trials. We define outliers as values falling outside the $[Q1 - 1.5 * \text{IQR}, Q3 + 1.5 * \text{IQR}]$ quartile interval. The chosen bin size is on the order of $40$ms for whole-trial period, and $10$ms for the Open-loop period.

After the firing rates are obtained, the next step in processing depends on the nature of the data analyzed, and involves using PCA. If we consider trial-level modeling, then perform PCA to project the firing rates onto the top PCs ($M = 50$ PCs, over 90% variance explained). If we consider session-level modeling, we first average the firing rates from 200ms before to the stimulus onset to the reward, then perform PCA to project the per-trial activity onto the top PCs from the concatenated trials ($M = 10$ PCs, over 85% variance explained).

Finally set our internal states $\boldsymbol{x}_n$ to be of dimension $D = 4$ for per-trial neural activity, and $D = 2$ for per-trial-averaged activity.

**Ephys data details**  We use Neuropixels recordings tied to the Brainwide data release Laboratory (2022). The results in this paper pertain to the first available session, with session id `ae8787b1-4229-4d56-b0c2-566b61a25b77`. We refer the reader to the data release website to obtain more information on the specific probe: `https://viz.internationalbrainlab.org/app`.

**Data**  we use a 4:1 train to test split for the per-trial spike train analysis. 20% of the trials are randomly selected as test trials – these trials span the whole session.

## C  Modeling

### C.1  Pólya-gamma augmentation for Gibbs sampling in infinite recurrent SLDS

We provide more details on inference challenges in an infinite recurrent SLDS model naively combining the dist-CRP and the SLDS. Following previous methodologies, we could use Gibbs sampling and leverage message passing to perform inference. It would require us to handle the conditional density

$$p(\boldsymbol{x}_{1:N}|\boldsymbol{c}_{1:N}, \boldsymbol{z}_{1:N}, \{\boldsymbol{y}_{1:N}, \boldsymbol{t}_{1:N}\}) \propto \prod_{n=1}^{N} \psi(\boldsymbol{x}_{n-1}, \boldsymbol{x}_n, z_n)\psi(\boldsymbol{x}_{n-1}, c_n, \boldsymbol{t}_{:n})\psi(\boldsymbol{x}_n, \boldsymbol{y}_n)$$

where $\psi(\boldsymbol{x}_n, \boldsymbol{x}_{n+1}, c_{n+1})$ is the potential from the continuous recurrent dynamics, and $\psi(\boldsymbol{x}_n, y_n)$ the evidence potentials. The recurrent connections introduce the dependencies captured in $\psi(\boldsymbol{x}_n, c_{n+1})$, which adds significant challenges for inference. Without it, in the standard SLDS, the potentials are all Gaussian, allowing for analytical integration.

Following Linderman et al. (2016), we can leverage Pólya-gamma augmentation (Polson et al., 2013; Linderman et al., 2015) to deal with these non-Gaussian factors. The key is that instead of performing a categorical choice over a pre-determined set of $K$ dynamical modes, we perform an association, a categorical choice, with a previous time-step $j \in \{1, \ldots, t\}$. Because of the conceptual categorical similarities in updating, we find similarities in inference methodology with the rSLDS. This is where

the choice of decay function $f$ comes in, in enforcing that link. This non-Gaussian factor is

$$\psi(\boldsymbol{x}_{n-1}, c_n, \boldsymbol{t}_{:n}) = p(c_n | \boldsymbol{x}_{n-1}, \boldsymbol{t}_{:n}) \propto \prod_{j=1}^{n} f(t_n - t_j; \beta(\boldsymbol{x}_{n-1}))^{\mathbb{I}[c_n=j]} \alpha^{\mathbb{I}[c_n=n]}$$

$$= \alpha^{\mathbb{I}[c_n=n]} \prod_{j=1}^{n-1} \left( \frac{e^{[\boldsymbol{\nu}_n]_j}}{1 + e^{[\boldsymbol{\nu}_n]_j}} \right)^{\mathbb{I}[c_n=j]}$$

for $\boldsymbol{\nu}_n \in \mathbb{R}^{n-1}$, $[\boldsymbol{\nu}_n]_j := \beta(\boldsymbol{x}_{n-1}) \cdot (t_n - t_j)$. We can leverage the following integral quantity

$$\frac{(e^\nu)^a}{(1 + e^\nu)^b} = 2^{-b} e^{\kappa\nu} \int_0^\infty e^{-\omega\nu^2/2} p_{\mathrm{PG}}(\omega|b, 0) \mathrm{d}\omega \qquad b > 0, \kappa = a - \frac{b}{2}$$

to introduce auxiliary variables $\{\omega_j\}_{j=1}^n$ such that the conditional density $p(c_{n+1} | \boldsymbol{x}_n, \boldsymbol{t}_{:n+1}, \omega_n)$ becomes Gaussian

$$\psi(\boldsymbol{x}_n, c_{n+1}, \boldsymbol{t}_{:n+1}, \omega_n) \propto \alpha \, \mathcal{N}(\boldsymbol{\nu}_n \mid \Omega_n^{-1} \boldsymbol{\kappa}_n, \Omega_n^{-1}) \tag{11}$$

where $\Omega_n = \mathrm{diag}(\omega_{1:t-1})$, and $[\boldsymbol{\kappa}_n]_j = \frac{1}{2}\mathbb{I}[c_n = j]$, $\boldsymbol{\kappa}_n \in \mathbb{R}^n$. With this augmentation, the required potentials are Gaussian and the integral can be calculated analytically. We refer the reader to Blei & Frazier (2011) for subsequent details on how one handles the messages $m_{n \to n+1}(c_{n+1})$.

While we do obtain an analytical expression for the augmented potentials in (eq. 11), such potentials are multivariate Gaussians of size $n$ for each time-step $n \in \{1, \ldots, N\}$. This adds significant computational complexity, both in handling the potentials and in any required marginalization.

## C.2 IMPLEMENTATION OF THE IRSLDS

In this appendix subsection, we present code modules and functions relevant to the modeling.

**Parallel scan** We cast the linear dynamics of the transition matrices $W_n$ as a scan operation to efficiently parallelize the computation. The SSM package relies on Numba and autograd, the latter not having a readily implemented associate scan operation. To this end, we implemented in Numba and Numpy the scan operation, using numba for the parallization (Listing 1). Unfortunately, this scan operation cannot leverage the automatic differentiation from autograd. We found however that including the scan in the computation does not significantly change the learned solutions in the irSLDS parameters. Thus in practice, we use the scan to do faster hyperparameter search or gradient-free computation (e.g. generating samples, evaluating likelihoods), but rely on the full sequential version for the final training.

```
1  @numba.jit()
2  def binary_operator(q_i, q_j):
3      r""" Binary operator for parallel scan of linear recurrence.
4          Args:
5              q_i: tuple containing U_i and b_i at position i        (K,K,), (K,K,)
6              q_j: tuple containing U_j and b_j at position j        (K,K,), (K,K,)
7          Returns:
8              new element ( U_out, bias_out )
9      """
10     U_i, b_i = q_i
11     U_j, b_j = q_j
12     return np.matmul(U_i, U_j), np.matmul(b_i, U_j) + b_j
13
14 @numba.jit(parallel=True, forceobj=True)
15 def apply_scan(U, bias_elements):
16     r"""
17     Apply parallel scan for a length T linear recurrence of the form
18
19         x[0] = bias_elements[0]
20         x[t] = U x[t-1] + bias_elements[t]
21
22     Args:
23         U: Recurrent matrix, np.ndarray        (K,K,)
24         bias_elements: Additive bias elements, np.ndarray        (T+1,K,)
25             First bias element is the initialization for x.
26     Returns:
27         X: State dynamics, np.ndarray        (T,K,)
28     """
29     T = len(bias_elements) - 1
30     K = U.shape[0]
31
32     # Initalization
33     X = np.empty((T, K, K), dtype=np.float64)
34     q_i = ( np.eye(K), np.zeros_like(bias_elements[0]) )
```

```
35
36     # Use numba prange for parallization
37     for i in numba.prange(T):
38         q_i = binary_operator(q_i, c_i)
39         X[i] = q_i[1]
40     return X[1:]
```

Listing 1: `Numba` implementation of associative scan

**Transition module**  We implement the irSLDS within the `SSM` package from Linderman et al. (2020). This can be accomplished by using the same SLDS parent class as the (r)SLDS models, and specifying a new discrete state process. We implement this discrete state process as a `ssm.transitions.Transitions` module.

```
1  class HeatRecurrentTransition(Transitions):
2      """
3      Use w_t as a probability dist to sample z_{t+1} from.
4      w_t follows the heat equation: w_t \in L^1 satisfying
5            D_t w_t = \gamma D_xx w_t
6      """
7      def __init__(self, K, D, M=0, gamma=1.0, kappa=0.4, scan=False):
8          super().__init__(K, D, M)
9
10         # Parameters linking past state to current state distribution
11         self.beta = beta
12         self.scan = scan
13         self._set_beta()
14         self._set_U() # FDM matrix for heat equation
15         self.kappa = kappa
16
17         # Parameters linking past observations to state distribution
18         self.Vs = npr.randn(K, M)               # Inputs encoding, per state
19         self.Rs = npr.randn(K, D)               # Previous observations encoding, per state
20         self.r = npr.randn(K)
21
22     def _set_beta(self):
23         delta_t = 1.0                           # assume regular emissions.
24         delta_x = np.sqrt((8 * self.gamma * delta_t))    # determine the delta x such that FMD is stable
25         assert delta_t <= (delta_x ** 2)/(4 * self.gamma) # general assertion for stability
26         self.beta = (self.gamma * delta_t) / (delta_x ** 2)
27
28     def _set_U(self):
29         self.U = self.beta*np.diag(np.ones(self.K-1), k=1) +\
30                  (1-2*self.beta)*np.diag(np.ones(self.K)) +\
31                  self.beta*np.diag(np.ones(self.K-1), k=-1)
32         if self.mod:
33             self.U[0,-1] = self.beta
34             self.U[-1,0] = self.beta
35
36     def get_sufficient_statistics(self, data, input, mask, tag, w_state=None, SCAN=False):
37         T, _ = data.shape
38
39         # Initialization
40         ws = []
41         if (w_state is None): # Init
42             wt = 1/self.K * np.ones((self.K, self.K))
43
44
45         # Dynamics
46         if SCAN:
47             # Use einsum to parallelize over time the biases encoding
48             kappa_U = np.einsum('ij,jk->ik', self.U, self.kappa*np.eye(self.K))
49             bias_elements = np.repeat(kappa_U[np.newaxis, :, :], T, axis=0) + \
50                 np.repeat(np.einsum('tm,mk->tk', input, self.Vs.T)[:, np.newaxis, :], self.K, axis=1) + \
51                 np.repeat(np.einsum('td,dk->tk', data, self.Rs.T)[:, np.newaxis, :], self.K, axis=1)
52
53             # Add initial condition to initial bias
54             bias_elements = np.concatenate([wt[np.newaxis, :, :], bias_elements], axis=0)
55
56             # Use the scan operation to parallelize the linear dynamics over time
57             ws = apply_scan(self.U, bias_elements)
58             return ws
59         else:
60             # Use sequential dynamics computation
61             for t in np.arange(T-1):
62                 wt = np.dot(wt, self.U.T)                              # FDM
63                 wt = wt + self.kappa*np.eye(self.K)                    # self reinforcement.
64                 wt = wt + np.dot(input[t], self.Vs.T) + np.dot(data[t], self.Rs.T)   # inputs
65                 ws.append(wt)
66             ws = np.array(ws)
67             return ws
68
69     def log_transition_matrices(self, data, input, mask, tag, w_state=None):
70         ws = self.get_sufficient_statistics(data, input, mask, tag, w_state=w_state, SCAN=self.scan)
71         normalized_log_Ps = ws - logsumexp(ws, axis=2, keepdims=True)
72         if self.alpha>0.:
73             reweighted_log_Ps = (1-self.alpha)*normalized_log_Ps
74             out = reweighted_log_Ps - logsumexp(reweighted_log_Ps, axis=2, keepdims=True)
```

```
75          else:
76              out = normalized_log_Ps
77          return out
```

Listing 2: Basic implementation of the transition matrix dynamics with heat-equation prior

### C.3   TRANSITION MATRIX DYNAMICS

We explicit here the dynamics of the transition matrix $W_n \in \mathbb{R}^{K \times K}$. The equation for the sufficient statistic $\boldsymbol{w}_n$ dynamics in (8), which we rewrite here for reference,

$$\boldsymbol{w}_{n+1} \mid \{\boldsymbol{w}_n, \boldsymbol{x}_n, z_n = i\} = U\boldsymbol{w}_n + \kappa\mathbf{1}_i + R\boldsymbol{x}_n + r,$$

refers to row $i$ of the transition matrix $W_{n+1}$, so that

$$[W_{n+1}]_{i,j} = [\boldsymbol{w}_{n+1} \mid \{\boldsymbol{x}_n, z_n = i\}]_j . \tag{12}$$

We note that all vectors are columns vectors.

To get the dynamics of the transition matrix itself, we must expand out the computation for each row. Specifically, given $W_n$, we obtain

$$
\begin{aligned}
W_{n+1} &= \begin{bmatrix} \left(U[W_n]_{1,:}^\top\right)^\top \\ \vdots \\ \left(U[W_n]_{K,:}^\top\right)^\top \end{bmatrix} + \begin{bmatrix} \kappa\mathbf{1}_1^\top \\ \vdots \\ \kappa\mathbf{1}_K^\top \end{bmatrix} + \begin{bmatrix} (R\boldsymbol{x}_n + r)^\top \\ \vdots \\ (R\boldsymbol{x}_n + r)^\top \end{bmatrix} \\
&= \begin{bmatrix} [W_n]_{1,:}U^\top \\ \vdots \\ [W_n]_{K,:}U^\top \end{bmatrix} + \kappa\mathrm{Id} + (R\boldsymbol{x}_n + r)^\top \otimes \mathbf{1} \\
&= W_n U^\top + \kappa\mathrm{Id} + (R\boldsymbol{x}_n + r)^\top \otimes \mathbf{1} \tag{13}
\end{aligned}
$$

as the dynamics of the transition matrix. One can find these dynamics in our implementation of the transition module in Listing 2, lines 62–64. Unless otherwise noted, we use $\kappa = 0.4$, and $\beta = 1.0$ for defining $U$.

# D RESULTS

## D.1 ACCURACY METRICS

All models were trained for 100 iterations, for every task. We only report the "mean" test log-likelihood, i.e. the log of the mean of the test likelihoods.

Table 2: *Performance metrics on NASCAR experiments–Expanded.* We report the marginal log-likelihood on test data, evaluated with Sequential Monte Carlo. Test data-sets consist of $T = 200$ excluded steps in NASCAR, and excluded trajectory of $T = 1000$ steps in extended NASCAR. We also report the dynamical mean-square error (MSE) between the sample $x$ state-space flow and true flow field. We first reflect and rotate the sample trajectories to best align them with the flow field.

| Model | $K$ | # parameters | Test log-likelihood (↑) (nats) | Dynamical MSE (↓) Mean | IQR |
|---|---|---|---|---|---|
| **NASCAR** | | | | | |
| SLDS | 4 | 92 | 1331.35 | 0.930 | $[0.247, 0.629]$ |
| | 8 | 176 | 1592.81 | 305.046 | $[0.058, 0.759]$ |
| rSLDS | 4 | 100 | 1628.57 | 0.010 | $[0.005, 0.015]$ |
| | 8 | 192 | 1613.53 | 0.011 | $[0.010, 0.013]$ |
| irSLDS | 4 | 88 | 1656.10 | 0.015 | $[0.007, 0.015]$ |
| | 8 | 136 | 1611.45 | 0.011 | $[0.006, 0.014]$ |
| **Extended NASCAR** | | | | | |
| SLDS | 6 | 130 | 7353.22 | 316.75 | $[0.29, 5.46]$ |
| rSLDS | 6 | 142 | 8015.93 | 0.35 | $[0.04, 0.48]$ |
| irSLDS | 6 | 112 | 8219.41 | 0.24 | $[0.10, 0.27]$ |

Table 3: *Performance metrics on IBL experiments*: Marginal Log-Likelihood of held-out test data, evaluated with Sequential Monte Carlo. Higher is better.

| | **Per trial** Test log-likelihood (↑) | | | **Per session** ELBO (↑) |
|---|---|---|---|---|
| Model | $K = 2$ | $K = 4$ | $K = 8$ | $K = 8$ |
| rSLDS | $-1469.759$ | $-1437.519$ | $-1399.030$ | $-6.67 \times 10^3$ |
| irSLDS | $-1446.692$ | $-1461.777$ | $-1392.508$ | $-6.69 \times 10^3$ |

## D.2 EMPIRICAL RUNNING TIMES

Table 4: Empirical running times, in seconds per EM iteration, reported for the standard NASCAR task and trial-level IBL. The experiments were ran on an AMD EPYC 7H12 64-Core Processor, on 10 cores. No GPU was used.

| (s/iter) Model | NASCAR $K = 4$ | Per trial IBL $K = 2$ | $K = 4$ | $K = 8$ |
|---|---|---|---|---|
| SLDS | $0.13 \pm 0.02$ | $6.02 \pm 0.06$ | $12.9 \pm 0.2$ | $8.7 \pm 0.3$ |
| rSLDS | $0.18 \pm 0.03$ | $7.32 \pm 0.10$ | $12.3 \pm 0.6$ | $18.2 \pm 0.3$ |
| irSLDS | $3.21 \pm 0.71$ | $48.47 \pm 2.91$ | $56.4 \pm 3.1$ | $70.0 \pm 5.3$ |
| irSLDS (w/ scan) | $0.54 \pm 0.08$ | $11.73 \pm 0.50$ | $14.0 \pm 0.8$ | $18.4 \pm 1.0$ |

## D.3 HYPER-PARAMETER INFLUENCE

**Scaling the number of states** $K$    We include results in Fig. 5 on the impact of scaling $K$ across models. We consider the standard NASCAR task, and evaluate model performance on $T = 200$ held-out time steps, averaged 5 random initialization seeds, and report mean and standard deviation.

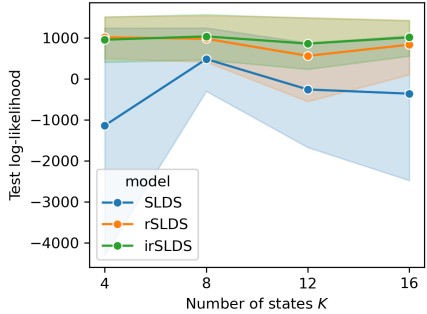

Figure 5: Impact of scaling $K$ on model performance. Test log-likelihood on NASCAR task as a function of the hyperparameter $K$. See text for details.

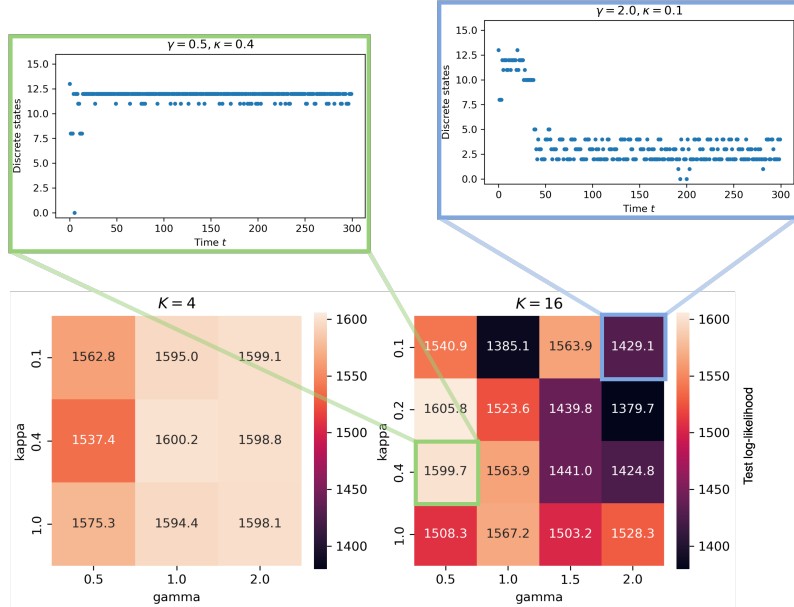

Figure 6: Impact of the hyperparameters $\{\gamma, \kappa\}$ on the irSLDS model. We report test log-likelihood on NASCAR task as a function of the hyperparameters, and plot example discrete state trajectories. See text for details.

We trained and tested models using $K \in \{4, 8, 12, 16\}$ states, thus considering the over-parametrized regime over the true $K = 4$ states for the NASCAR task. We observe that the irSLDS maintains a high performance across seeds, matched by and slightly over-performing the true rSLDS model class. The SLDS model class exhibits more variability. Note that in contrast to Table 1, we generated new train/test datasets for each seed.

**Heat equation parameters in the irSLDS**  We consider the impact on performance of the self-reinforcement parameter $\kappa \in \mathbb{R}$ and the parameter of decay $\gamma > 0$ in the heat equation prior of eq. equation 8 in §3.3. For all experiments and results reported throughout the paper, we chose $\gamma = 1.0$ and $\kappa = 0.4$, we let $\Delta t = 1.0$, and we set $\Delta j = \sqrt{8\gamma\Delta t} = 2\sqrt{2}$ (for stability). We report in Fig. 6 test log-likelihood for the standard NASCAR task, on one random seed, over a grid of hyper-parameter $\{\gamma, \kappa\}$ combinations. In terms of performance, we observe that the specific choice of hyper parameters has minimal effect for a low number of states $K = 4$. The impact becomes more evident when increased to $K = 16$, as a consequence of the more fine grained discrete state space support for the heat equation PDE. We plot prior sample trajectories, with $\alpha = 0.$, for two exemplar combination of parameters to illustrate their influence.

