# OpenReview forum: "Parsing neural dynamics with infinite recurrent switching linear dynamical systems"
_ICLR.cc/2024/Conference — ICLR 2024 poster_

### Official Review · Reviewer_k1CY · 2023-10-28

**Soundness:** 4 excellent
**Presentation:** 3 good
**Contribution:** 3 good
**Rating:** 6
**Confidence:** 5

**Summary:**

This work extends the rSLDS. It embeds the latent modes (discrete states) in Euclidean space with PDE, and endows the former nominal states with geometry. The proposed method was validated on synthetic data and real neural recordings. The results show that the proposed method better captures the change of dynamics on nonstationary DS.

**Strengths:**

This work shows good performance for nonstationary systems.
The extension brings up new interpretations on the discrete states.

**Weaknesses:**

rSLDS has the intuition of partitioning the latent space into regions that are governed by different linear DS individually, and the resultant nonlinear DS globally. It's not very suitable to do comparison on a nonstationary system.
The goodness of fit numbers do not show irSLDS significantly outperform rSLDS. It seems better state inference does not help much with explaining the data.

**Questions:**

In the extended NASCAR example
- Why the trajectory is not smooth? The bias is too large?
- Why not show MSE for extended NASCAR in Tab 1.

It would be more convincing to see the discrete state inference from continuous recording for real nonstationary system.

---

> ### Author Response · Authors · 2023-11-19
>
> Thank you for your review!  We hope what follows answers your remaining questions.
>
> **Weaknesses**: We start by first acknowledging the weakness raised. Indeed, the partitioning into different linear DS you mentioned is an important intuition. It is present in our model through the recurrent ($x_n \to z_{n+1}$) connections, but with the effective partition being time-varying. For rSLDS it is indeed a fixed global nonlinear DS, which induces a weaker prior over non-stationary dynamics. Nonetheless, the resulting system can still support non-stationary dynamics, which we believe makes comparisons on non-stationary systems still justified. A simple extension to the rSLDS that would allow for better modeling of nonstationarity would be to include an additional dependence on time $n$ through external inputs. This way, this partition over the $\boldsymbol x$ state space would be allowed to also vary with time. If deemed relevant, we would be happy to include this extension analysis as a supplement in the final version!
>
> **Questions**: Addressing your questions:
> - Extended NASCAR: We assume you are referring to the sample continuous states in Figure 2E. The posterior mean trajectories in Figure 2C are smooth, but in Figure 2E we plot the *sampled* continuous state trajectories, that will by definition be noisy.
> - Thank you for pointing this out. We have now updated Table 1 in the main text to incorporate the dynamical MSE, as well as Table 2 in the Appendix.
>
> Finally, could you expand on this sentence?
> > It would be more convincing to see the discrete state inference from continuous recording for real nonstationary system.
>
> Thank you again for your feedback. Let us know if we can better answer any remaining questions you may have!

---

> > ### Comment · Reviewer_k1CY · 2023-11-20
> > **Response to the authors**
> >
> > Thank you for addressing my concerns.
> >
> > > It would be more convincing to see the discrete state inference from continuous recording for real nonstationary system.
> > In the electrophysiological recording example, the model was fit to trials if I understand it correctly so that you lost the continuity of state $x$ and $z$ between the trials. So, I wondered whether the method would obtain the same inference especially of the discrete state if you fit it to the continuous recording. Certainly, it doesn't have to be done on this dataset for practical considerations.

---

> ### Author Response · Authors · 2023-11-21
>
> We are glad to read our reply addressed your concerns!
>
> Thank you for clarifying. Indeed, there are considerable practical considerations to the full continuous recordings since we'd be dealing with an order of magnitude longer sequences. There are also considerations on the modeling side, as modeling the continuity in x and z *between* trials has to be handled carefully. For instance, in the decision-making data considered here, there are inter-trial breaks (elapsed times) that might make the latent states discontinuous. To tackle this, one approach is to define different dynamical transitions for the latents at those time steps from one trial to another (see e.g. (Roy et al, Neuron, 2021)). Thank you for raising this point, we will include a comment on this aspect in the limitations section of the paper.

---

### Official Review · Reviewer_Rx1k · 2023-11-04

**Soundness:** 3 good
**Presentation:** 3 good
**Contribution:** 3 good
**Rating:** 6
**Confidence:** 3

**Summary:**

This paper builds upon the recurrent switching linear dynamical system (rSLDS) in the context of modeling latent dynamics of neural spike train data. The infinite recurrent SLDS (irSLDS) is proposed to overcome the fixed number of discrete states in the rSLDS, which can limit its capabilities on trial-varying data that may require a varying number of states. Prior methods have proposed using Dirichlet processes to infer the number of states from the data, often using a Chinese Restaurant Process (CRP). But this is limited in the rSLDS case, since the CRP has exchangeability assumptions which the state-dependence of the rSLDS breaks. The proposed method introduces a sufficient statistic to express the cluster allocations of a distance-dependent Chinese Restaurant Process (dist-CRP) which can be used to address the exchangeability issue and determine the discrete state switching probabilities. This sufficient statistic is implemented following a heat-equation combined with finite difference methods, resulting in linear recurrences. The proposed irSLDS is compared with SLDS and rSLDS on a synthetic NASCAR experiment and compared to the rSLDS on Neuropixel data recorded from mice.

**Strengths:**

- For the most part, the paper is clear and presented well

- The method is motivated well. In the scientific community, there is a strong desire for models that are both expressive and interpretable that can be used to gain insights into data, e.g. neural spike trains.  While rSLDS has been popular in this regard, addressing its weakness of having a fixed number of states is of interest to the community

- The idea to characterize the distance-dependent CRP in the context of SLDS with a heat equation appears (to my knowledge) to be novel and is an interesting idea that seems to lead to a relatively straightforward implementation (the basic implementation of the code in the appendix is also appreciated).

- The considered experiments appear to show that the irSLDS can have a performance/interpretability edge in practice and can potentially lead to insights that rSLDS cannot

**Weaknesses:**

- The paper would benefit from providing a bit more context/intuition behind the Polya-gamma approach in Section 3.1 and its weaknesses. Just reading this section currently does not make it as clear why the approach in section 3.2 is necessary. Section C.1 in the appendix is helpful. Perhaps a bit more details about the Gaussians with time-dependent sizes could be added to the main paper to make this point more clear?

- The experimental results are the biggest weakness of the paper. It is not currently obvious that a practitioner should clearly reach for irSLDS as the preferred method.
  -  Are there other datasets (synthetic or real) that could be added to make the benefits of irSLDS more clear?
   - Can the extended NASCAR example be made more extreme (e.g. more states) to make the differences between irSLDS and rSDLS more apparent? It is not clear that the differences in test log-likelihood are that drastic (and there do not appear to be underlying states of rSDLS displayed to compare to)
   - Why not also show rSLDS continuous states in Figure 2.E? I would expect these to not look good (unlike in Figure 2.D where rSLDS and irSLDS look similar) and think highlighting this would be helpful

- It is claimed in Section 3.3 that irSLDS has a similar time complexity to rSLDS. It would be nice to see some kind of empirical runtime comparison to support this. How does this change when choosing different upper bounds of $K$?

- Related to the 2 previous points above, it seems 2 potentially simple baselines would be an rSLDS with a large number of states or alternatively performing a grid search with rSLDS over different state sizes. On the other hand, given the flexibility of irSLDS, I would think it should be able to achieve better performance than these 2 baselines in one-shot. Instead, it seems the same amount of grid search was done for each method over the number of states. Alternatively, if either of these two baselines still takes less time/cost than running irSLDS (this is where the empirical runtime comparison comes in), then it seems perhaps rSLDS would be preferable for a practitioner in many cases? I think empirical evidence to resolve these doubts, both in terms of the empirical runtime under different scenarios and the time/cost per methods to achieve strong results, would strengthen the paper.

**Questions:**

For major points, see weaknesses above.

Additional questions/comments:

- In Table 1, can you make it more clear what the true number of states in the system are and what size $K$ is used by the models? I believe the number currently listed is the true number of states in the system, but it is unclear what $K$ was chosen for the models.

- Unless I am missing something, I believe the last paragraph of Section 3.3 has an error regarding the computational complexity of the sequential vs parallel versions of the algorithm. It is stated that the sequential scan requires $\mathcal{O}(K^3T)$ ops, but shouldn't this be  $\mathcal{O}(K^2T)$ ops? This is because it requires $T$ sequential matrix-vector multiplications (compared to the cubic cost of matrix-matrix multiplications of the parallel version of the scan). Thus the choice of parallel vs sequential scan would be dependent on the state size and sequence length, since the cubic cost could grow quite large for larger state sizes.

- This paper https://arxiv.org/abs/2111.01256 seems relevant to infinite switching states and could be cited as related work

---

> ### Author Response · Authors · 2023-11-19
>
> Thank you for your careful review! We appreciate the time and dedication towards highlighting the strengths and valuable feedback to make the paper better. We hope below to have summarised faithfully the weaknesses and questions you raised.
>
> First, addressing the weaknesses point by point:
> - **Polya-gamma**: Thank you for the suggestion on the Polya-gamma augmentation, we will add some of the suggested details on the time-dependent sizes in the main text for the final version.
> - **Extended NASCAR**: The rSLDS states equivalent to Figure 2E-F can now be found in Appendix Figure 5 -- we apologize for their omission in the original submission. Indeed as you predicted, the sampled continuous states do not look as good for the rSLDS, whereas the samples are much more in line with the posterior mean for the irSLDS. Furthermore, we also provide the equivalent to Figure 2F for the rSLDS by plotting the most likely $\hat z_{1:T}$ trajectory, and we see that the rSLDS model mistakenly maintains only $K=4$ states after the switch to the alternate dynamics (where $K=6$). Together these results reinforce the performance of the irSLDS over the rSLDS in this non-stationary case. We can include these details in the main text for the final version.
>
> - **Empirical considerations**: Finally, thank you for bringing up concerns on the practitioner, empirical, side. First, we hope the provided running times in the updated manuscript (see Appendix D.2.) help give a better sense of the computational cost. They support, for instance, how a grid search over $K$ for the rSLDS would be quite computationally expensive compared to our irSLDS, which has more flexibility with respect to the inferred $K$. Furthermore, recall that we provided empirical evidence (in Section 4.2 on the IBL data) towards the irSLDS model's ability to uncover fluctuations in the number of discrete states $K$ on a trial-by-trial basis. The rSLDS provided a slightly worse fit, and did not uncover those fluctuations as much. This indicates that even if the grid-search over $K$ with the rSLDS *was* computationally efficient and preferred, it would still yield a constant $K$ that might miss those within-dataset fluctuations.
>
> As for the additional questions/comments:
> - **Computational complexity**: Thank you for pointing this out, this is highlights a good need for clarification! For the parallel scan, we focus on its use for the computation of the full transition matrices $W_{1:T}$. Since these are matrix dynamics, and not vector dynamics, they do require $\mathcal{O}(K^3 T)$ ops, from $T$ sequential matrix-to-matrix multiplications. We agree that the text was misleading, making it seem that we use it for the computation of the vector sufficient statistics $\boldsymbol w_n$. This latter computation only really arises if we want to generate samples from the model, in which case it is indeed sequential -- but so are all other models. We provided some clarification in the updated manuscript, and will pay attention to those details for the final version!
> - Thank you for the relevant reference, we've included it in the introduction.
>
> Thank you again for your feedback. We look forward to answering any further questions you have!

---

> > ### Comment · Reviewer_Rx1k · 2023-11-21
> >
> > Thank you for the response and clarifications. I have increased my score.

---

> > > ### Author Response · Authors · 2023-11-21
> > >
> > > We are glad to read this provided clarification. Thank you for taking the time to consider our replies and for being willing to re-evaluate the paper!

---

### Official Review · Reviewer_uvcW · 2023-11-05

**Soundness:** 4 excellent
**Presentation:** 4 excellent
**Contribution:** 3 good
**Rating:** 8
**Confidence:** 3

**Summary:**

The authors propose a method for fitting dynamical models onto neural data that estimates the latent state of the system/organism.  A commonly used approach in the recent years is to fit a recurrent switching linear model, that approximates the latent dynamics with a finite set of low dimensional linear systems, where the system state is assumed to switch at discrete points between these discrete states. However this formalism requires stationary dynamics across all trials and does not impose any geometry on the discrete latent state space. Here the authors propose an extension to the recurrent switching linear systems, by considering that the evolution between the discreet states follow a Markov process itself (a distance-dependent Chinese restaurant process). To perform inference on the proposed model, they introduce a time-dependent sufficient statistic for the discrete state, the influence function, that is assumed to follow the heat (PDE) equation. The advantage of this statistic is that it is deterministically determined given the latent discrete state and the observations, and therefore does not arise in the posterior inference. Thus the authors can apply existing variational Laplace-EM proposed for these type of models like in Zoltowski et al. (2020).

They validate their approach on a synthetic dataset devised to imitate o stationary discrete latent state dynamics, and on an neural electrophysiology dataset from IBL with neural recordings from mice performing  a decision-making task.

Overall the paper is very well written and is a sound extetion to the existing literature of switching linear dynamical systems.

**Strengths:**

- They endow the switching linear dynamical system framework with a latent discrete-state Markov structure that allows the continuous state of the system to guide the discrete state switches.
- The proposed formalism induces a geometry in the discrete latent state between switches.
- The paper is very well written providing a very well constructed introduction and build-up to their proposed extention.

**Weaknesses:**

- The resulting computations for this model are considerably more expensive compared to the previous approaches recurrent switching linear system framework, however this might be the tradeoff of dealing with non-stationarity.

**Questions:**

- In Figure 3 F lower,  irSLDS identifies the state 2, although it does predict (as I understand from the absence of orange dots) that the discrete state of the system was never at that state through the trials. Isn’t that strange? Shouldn’t the framework have estimated 7 states in total for this dataset?

- In the NASCAR experiment, can you provide the same plot as in Figure 2F that shows the sequence of the identified states also for the rSLDS?

- What are the limitations for considering Poisson instead of Gaussian observation process for the irSLDS framework?


Minor:
- Conclusion has various typos, i.e. first sentence of conclusion: “a”-> “an”, missing ’s’-es.
- I understand that in the caption of Figure 1 you do not want to write out the full abbreviated terms, but you can at least provide a hyperlink to their mention in the main text for the reader to follow.

---

> ### Author Response · Authors · 2023-11-19
>
> Thank you for your positive review and feedback! We hope what follows answers your remaining questions.
>
> To address the weakness raised, handling non-stationarity is indeed often challenging. Fortunately, the discrete-state process is defined by only a few key parameters, which helps in terms of interpretability and can help with hyperparameter optimization during training. Furthermore, our implementation of the scan operation does significantly reduce the computational complexity (see empirical running times added in Appendix D.2) where it can be deployed.
>
> As for your questions:
> - Indeed we set $K=8$ as an upper bound. For this specific visualization of inferred discrete states, namely the most likely sequence, indeed it does not use state #2. This state is still however well defined for the model, and other analyses (such as for instance smoothing marginal posteriors $p(z_t \mid \hat x_{1:T}, y_{1:T})$) might leverage that state.
> - Thank you for pointing that out! We provide the same plot as Figure 2E-F for the rSLDS in Figure 5 in the Appendix. As an overview, we observe that the sampled continuous states do not look as good for the rSLDS, whereas the samples are much more in line with the posterior mean for the irSLDS. As for the equivalent to Figure 2F for the rSLDS, with the most likely $\hat z_{1:T}$ trajectory, and we see that the rSLDS model mistakenly maintains only $K=4$ states after the switch to the alternate dynamics (where $K=6$). Together these results reinforce the performance of the irSLDS over the rSLDS in this non-stationary case. We can include these details in the main text for the final version.
> - There would be no limitations to considering Poisson instead of Gaussian observation process for the irSLDS framework, with this specific implementation. Indeed, we do not leverage the Gaussian observation process directly, which one could do for instance to do efficient (even exact if given the $z_{1:T}$ sequence) marginalization of the continuous states in SLDS models. Here, all of our implementation is based on the SSM package, so all computation is numerical and readily generalizable to other emission processes.
>
> Finally, thank you for pointing typos/clarifications, we will integrate them in the final version. We look forward to answering any further questions you have!

---

> > ### Comment · Reviewer_uvcW · 2023-11-22
> >
> > I have read the responses to my comments, and I would like to thank the authors for the clarifications!

---

### Official Review · Reviewer_u3HM · 2023-11-06

**Soundness:** 3 good
**Presentation:** 3 good
**Contribution:** 3 good
**Rating:** 8
**Confidence:** 2

**Summary:**

The authors proposed a novel infinite recurrent switching linear dynamical system (irSLDS) model, which combines partial differential equations (PDE) theory, semi-parametric formulation, and switching dynamical models. The authors validate and demonstrate the capabilities of their model on simulated data, and compare their model with existing popular methods. Finally, they use the model to explore mouse electrophysiological data during decision-making and uncover strong non-stationary processes underlying both within-trial and trial-averaged neural activity.

**Strengths:**

The authors introduced an influence function of the states in switching linear dynamical systems, which controls the switches between discrete states by adding time-dependent parameters to constrain the duration of a state, and a space smoothness prior to the influence function.

**Weaknesses:**

How is the smoothness prior determined? In the paper, the authors state “Each time a discrete state j is chosen, “nearby” states also become more probable at the next time step”, it may be better to show the comparison with no smoothness or other smoothness formulations.

The prior based on dist-CRP is sensitive to time, what if there is a state with a long duration in the real dynamics? Or what if the dynamics are influenced by time a lot, such as in a Lorenz attractor?

How was the upper bound of K in irSLDS determined? Since the method is called infinite rSLDS, can the K be very large?

Clarity:
In 4.2 paragraph3, “we fit a spline to this curve a ...” seems to have a typo.

**Questions:**

See 'Weaknesses'.

---

> ### Author Response · Authors · 2023-11-19
>
> We thank you for the review! Specifically, we appreciate bringing to light the influence of hyper-parameters. Addressing the weaknesses point by point:
>
> - **Smoothness**: Indeed influence from ``nearby'' states is mediated by the diffusion parameter $\gamma$, or in the FDM implementation, $\beta \propto \gamma$. For the results reported in the submission we fixed $\beta = 1.0$ (for all tasks) determined from a quick hyper-parameter search. We agree an exploration of smoothness, per task, would be insightful -- we propose to include the log-likelihood search as well as example generated trajectories in the final version.
>
> - **Time duration and dependence**: Similarly to the smoothness described above, discrete-state duration can be mediated by the $\kappa$ hyperparameter. We used $\kappa = 0.4$, determined from a quick hyper-parameter search. As with the smoothness, we could include in the final version details on this hyper-parameter search. Regarding the question raised on time dependence: this is where our accent on the inclusion of input-driven transitions, either from external inputs $\boldsymbol u_n$ or continuous internal states $\boldsymbol x_n$ (recurrence), can shine through. Indeed one could include the time $n$ as an external input, for instance by defining the sufficient statistic dynamics as
> \begin{equation}
>     \boldsymbol w_{n+1} = U \boldsymbol w_{n} + \kappa\mathbf{1}_{i} + R \boldsymbol x_n + r + g(n,i)
> \end{equation}
> for $g(n,i)$ some parametrized function (e.g. affine $Q^{(i)} n + q$). Concretely for the Lorentz system mentioned, this modeling could leverage the continuous latent $\boldsymbol x_n$ to dictate one discrete state per side of the attractor, or use the time dependence to do so.
>
> - **Number of states**: For the synthetic experiments, $K$ was set to the true known value ($K=4$ NASCAR, $K=6$ extended NASCAR). For the IBL task, we used $K=8$ and provided in Appendix section D results on the log-likelihood search over different values of $K$. To address your final question, we will provide in the final version an analysis for scaling $K$, showing for instance how the different models fare in ($K$) over-parametrized regimes.
>
> Thank you again for your feedback. We look forward to answering any further questions you have!

---

> > ### Comment · Reviewer_u3HM · 2023-11-20
> >
> > Thank you to the authors. I am raising my score to 7 in light of their responses.

---

> > > ### Author Response · Authors · 2023-11-21
> > >
> > > Thank you for taking the time to consider our replies and being willing to re-evaluate the paper!

---

### Official Review · Reviewer_ots3 · 2023-11-07

**Soundness:** 2 fair
**Presentation:** 2 fair
**Contribution:** 3 good
**Rating:** 5
**Confidence:** 3

**Summary:**

This work presents an addition to recurrent switching state-space models by using PDEs to develop an input-driven prior that induces state geometry over the HMM while still retaining its recurrent capability. The proposed infinite recurrent switching linear dynamical system (irSLDS) allows for more expressible and flexible state cardinality over a fixed number of discrete states while yielding fewer parameters compared to previous HMM models but greater performance on two time-varying datasets: NASCAR and the “Brainwide map” dataset from the International Brain Laboratory. This is achieved by differentiating the influence functions of each state using the heat equation as a spatial smoothness prior. The forward difference of the time derivative and central difference approximation to the second order spatial partial derivative are utilized to retrieve an infinite prior for the dynamics of the influence functions and sampled to compose the overall discrete state process.

**Strengths:**

The paper's originality derives from improvements it makes in the class of AR-HMM and SLDS models by removing exchangeability without removing input recurrence. This enables greater efficiency with expressibility. Overall, this work nicely integrates the heat-equation generalization and recurrent switching state space models. They present overall improvements on two different experiments with greater efficiency of parameters. Visuals are strong, particularly ones pertaining to the flow fields and the switching states of the rSLDS and irSLDS.

**Weaknesses:**

Most of the comparisons are done between rSLDS and irSLDS, yet the results are not convincing and substantially novel in the experiments conducted. If the authors are adamant about just comparing these two models, then it would be nice to see comparisons at greater scales instead of just testing 4-8 states for NASCAR and the IBL experiments. The results at just these scales do not appear novel.
Furthermore, both experiments show that the irSLDS models uncovered switching at task-relevant states, yet it isn't clear why adding the heat equation prior itself is the best choice in discovering this switching. The authors discuss their reasons for using the heat equations but don't explain further into infinite priors that possibly could have achieved the same if not more.
The efficient of the parallel scan is not clear. Other works that utilize scans to efficiently compute the transition probabilities such as S4D (Gu et al., 2022), DSS (Gupta et al., 2022), S5 (Smith et al., 2023) are done on diagonalized transition matrices. The connection to the parallelization of the linear dynamics in Equation 8 is unclear since the transition matrix in this work is not diagonalized, so this work does not reap the same computational benefits that they claim, and their algorithm cannot scale effectively since it incurs cubic cost.

**Questions:**

Smith et al. is cited in reference to the use of a scan operation to compute transition probabilities. However, S5 assumes a diagonal state matrix to efficiently compute the linear recurrence which takes a different form than the time-varying transition matrices in this work. How can this scan method scale efficiently?
Why does the heat equation prior directly enable discovery of greater fluctuations/switching in the data? Can you attribute the performance gains to this specific infinite prior?
Why not compare to other linear state space models and generalize outside of rSLDS and irSLDS that can allow for input-dependent transitions and recurrent relations without the explicit use of the influence functions as a geometric prior?

---

> ### Author Response · Authors · 2023-11-19
>
> Thank you for your thorough review, we appreciate the time and dedication towards making the paper better. We hope to have summarised faithfully the weaknesses and questions you raised, and we address them point-by-point below:
>
> - **Comparison with other models**: We agree that our analysis is limited to the switching linear dynamical system and its direct extensions. Our primary goal was to convey how a carefully designed prior (in our case the heat equation that captures the dist-CRP and induces a state geometry) with select hyperparameters can maintain similar performance/efficiency, while increasing interpretability. In line with this goal, we focus on only one model family, with and without this prior. We appreciate the feedback on reinforcing the analysis and will provide scalability results in the final version with a more thorough analysis beyond the 4-8 states!
>
> - **Heat equation prior**: We agree that there are alternative prior formulations, and we do not make claims that this is necessarily the *best* prior to uncover switching. These possible alternate priors are not discussed thoroughly in the current manuscript, and we will make sure to include a discussion element on that regard. Nonetheless, our claim is that this prior encapsulates important properties, and as such offers a good prior for model fitting and data-driven analysis. With regards to your question "Why does the heat equation prior directly enable discovery of greater fluctuations/switching in the data?": as mentioned in the text, we attribute this capability to the fact that the heat equation prior encapsulates the dist CRP (point #2, end of section 3.2), yielding a model with greater capacity to calibrate the number of states used from the data.
>
> - **Parallel scan**: Thank you for raising questions on this topic, we realize that there is a need for clarification! First, just so there is no doubt, we use the parallel scan operation for the computation of the transition matrices $W_n$, *not* for the computation of the dynamics of $\vec x_n$. Hence our dynamics matrix of study is the FDM matrix $U$, not the time-varying linear matrices for the continuous state dynamics. Now, indeed, the previous work mentioned using the scan operation has focused on dynamics with diagonal matrices, whereas our dynamics matrix is not (here, tri-diagonal). Still, the parallel scan operation can be deployed for any recurrent system with affine dynamics. The diagonalization mentioned is crucial for efficient computation as $K$ scales, but this is not our focus. Our main focus is in making the computation more efficient as $T$ scales, and the parallel scan allows that, by bringing the sequential $T$ to $(T/L + \log L)$ ops. In our updated manuscript, we make explicit the dynamics of $W_n$ in Appendix C.3 for reference, and provide empirical running times in Appendix D.2 to showcase the increased efficiency from the scan.
>
> Thank you again for your feedback. We look forward to answering any further questions you have!

---

### Author Response · Authors · 2023-11-19

We are grateful to all the reviewers for taking the time to read our paper and for providing valuable feedback!

We have updated the manuscript to include the various points raised so far. Here is a summary of the changes:
- We have included the rSLDS states in the extended NASCAR task, equivalent to Figure 2E-F, in Figure 5 in Appendix D;
- We have included empirical running times for the standard NASCAR task as well as per-trial IBL analysis in Table 4 in Appendix D.2., including the running time for the irSLDS with and without the scan operation;
- We added dynamical MSE results for the extended NASCAR task to Table 1 in the main text and Table 2 in Appendix D.1.;
- We corrected and clarified the notation in the scan operation section.

Finally, we fixed various typos and clarified sentences raised by reviewers.

---

### Meta-Review · Area_Chair_6D7e · 2023-12-10

**Metareview:**

The paper presents an extension to the recurrent switching linear dynamical system (rSLDS) model, introducing the infinite recurrent switching (irSLDS). This new model incorporates a semi-Markov discrete state process with latent geometry, leveraging PDEs for sufficient statistics to the discrete states. The irSLDS aims to address limitations of the rSLDS, such as fixed state numbers over trials and lack of latent structure. The authors validate their model with synthetic data and apply it to analyze mice electrophysiological data during decision-making, revealing strong non-stationary processes in neural activity.

Strengths:
- The approach is definitely novel.
- The application to real-world neural data is a strong point, demonstrating the model's potential in uncovering complex, non-stationary neural processes.
- The authors addressed a large portion of reviewers' concerns, during rebuttal.

Weaknesses:
- The paper predominantly compares irSLDS with rSLDS, potentially overlooking a broader comparison with other models in the domain.
- Although the model is tested on synthetic and real data, additional datasets or more varied conditions could strengthen the validation.
- Some reviewers pointed out concerns about the computational complexity and scalability of the proposed model, which might not have been fully addressed.

The majority of the reviewers' concerns were addressed by the authors. AC votes for the acceptance of the paper.

**Justification For Why Not Higher Score:**

The paper still has to get improved in comparing the results with a broader range of models in the domain.

**Justification For Why Not Lower Score:**

All major reviewers' concerns were addressed.

---

### Decision · Program_Chairs · 2024-01-16

Accept (poster)